# Statistical Process Control Using Control Charts with Variable Parameters

**Tadeusz Sałaciński *** , **Jarosław Chrzanowski and Tomasz Chmielewski**

Institute of Manufacturing Technologies, Faculty of Mechanical and Industrial Engineering, Warsaw University of Technology, 85 Narbutta Str., 02-524 Warsaw, Poland; jaroslaw.chrzanowski@pw.edu.pl (J.C.); tomasz.chmielewski@pw.edu.pl (T.C.)
* Correspondence: tadeusz.salacinski@pw.edu.pl; Tel.: +48-22-234-8243

**Abstract:** An extremely important issue in quality management is monitoring and diagnosing processes, and, subsequently, supervising them using so-called control charts. In typical production processes, charts with constant parameters are commonly used, such as x-R, x-s, CUSUM, EWMA and others, which, in most cases, are effective tools for process stability evaluation. Charts considered untypical (in statistical process control) are those with variable sample sizes, variable sampling intervals and/or variable control limits. Such charts are used when process analysis based on standard, well-known charts may lead to serious errors. Modern control charts are a response to the requirements of Industry 4.0 and are an excellent tool for supervising production processes. Their use together with Cp and Cpk indices and other process capability indices is a starting point for process improvement. The methodology of nonstandard charts is inadequately recognized and rarely used in practice. The theory of their design and examples of their use will be presented and characterized in this paper.

**Keywords:** statistical process control (SPC); control charts; process stability

## 1. Introduction

A very important problem in quality management is to monitor and diagnose processes, that is, to supervise them using so-called control charts. For typical production processes, commonly used control charts of constant parameters are, for example, x-R, x-s, CUSUM, EWMA and others, which in most situations are effective tools for process stability evaluation [1–3].

Quality engineering is a science based on a process and system approach, which, apart from being oriented towards the customer, includes basic quality management rules. Every enterprise is a specific system of interconnected areas and activities (processes) creating a compact logical entity. A process is an activity, or a set of activities, defined in an instruction or procedure, transforming the entry state (input factors) into the final state (output factors), including a feedback loop consisting of control variables that allow us to correct the process. Disturbances influence the process negatively. They may be random (so-called natural process variability or "noise") or systematic, which are usually possible to remove by changing the values of process control variables.

A very important aspect in the area of SPC is technological process stability and process capability analysis, as well as process supervision (monitoring, diagnosing and regulation) with control charts and capability indices. These very convenient tools are widely used today; nevertheless, when selecting a particular chart or factor type, many errors are made. Nowadays, it is not enough to use the simplest chart types. Technological process capability and very high process accuracy often demand the use of more sophisticated tools, such as new-generation control charts or dynamic charts (adaptive charts). They take into account variable process conditions and adapt to them by changing collected sample sizes, sampling periods or control limit locations.

A control chart presents measurement data from a production process, usually as means of low quantity samples plotted against time with a central line presenting the general mean value of these data. Additionally, on the chart control, lines are plotted with an area of natural process variability between them. They are plotted in $\pm 3$ s distance (three mean values of standard deviation of samples) from the central line. Observations from a stable (statistically regulated) process stay within the control limits. The appearance of an observation value beyond the stability area (self-variability area) is a signal. It is also a signal if observation values group in one of a few specific configurations, although none of the values are beyond the stability area (this is called a serial out-of-control signal). The appearance of a process out-of-control signal forces the process owner to take preventive and corrective measures.

The first control chart for mass production process control for electronic components was created in 1924 by W.A. Shewhart. Production engineers quickly realized what benefits come from the use of this tool. A process control chart (or, more precisely, a process regulation chart or a chart for process control) is used for process supervision by monitoring characteristic process values to identify deviations quickly, and in this way avoid further defective products being made. Such a situation may take place, for instance, in case of a machine getting out of control or in the case of its bad setup by the operator. This leads to the appearance of a systematic (special) process influence, which is signaled by the appearance of points beyond control limits on the control chart. The control chart is thus is intended for visualization of process dispersion and regulation rates.

A very important element in chart keeping is not only the quantity of measurement samples (the bigger it is, the more reliable the obtained results are), but also the frequency of their collection, which should be higher during a process's introduction into production. However, it should be remembered that numerous samples collected very often may increase final process costs significantly.

The reasons for process stability loss should be archived. This way, process "history" is built, which may be used in further preventing activities. With such process "history", reasons for stability loss are identified, which may be used in the Pareto analysis. Such analysis allows us to identify some groups of the most frequent reasons, which may then be prevented in advance. The elimination of these reasons may be connected to large investments (e.g., purchase of new machine tools, chucks, cutting tools), but it will significantly increase process precision and finally decrease its costs by an increase in competitiveness and customer satisfaction.

Nonstandard charts are those (in statistical process control) which use variable numbers of samples, variable periods of sampling and/or variable control limits. Such charts are used when process analysis based on standard, well-known charts may lead to serious errors. Contemporary techniques of nonstandard charts are inadequately recognized and not used in practice; therefore, in this paper, the theory of the design and use of such charts will be characterized.

Essentials of constant parameters control chart design are presented in works by Montgomery [4], Smith [5] and Rauwendal [6]. Classic control charts assume a normal distribution of measurement results. Some papers analyze processes in which normal distribution is not examined [7–10].

The literature on control charts includes only a few papers discussing charts with variable parameters [11–16]. The goal of future research is to elaborate the design of versatile and universal control charts with variable parameters using general four-parameter Burr distribution which allows us to describe distributions other than the normal Gauss one.

## 2. The State of the Art in the Field of Control Charts

### 2.1. Control Charts with a Variable Number of Samples

In contemporary nomenclature, such charts are so-called adaptive charts in which a variable number of samples causes a change of location of control limits. The current idea of adaptive charts should be broadened with charts with variable sampling intervals

and asymmetric (so, variable as well) control limits. These three kinds of charts will be described in the next three sections. The architecture of a control chart with a variable number of samples is based on the following rule: Let us assume that in constant time intervals samples of constant quantity $n_0$ are collected. Two additional quantities $n_1$ and $n_2$ are then determined, of which the first one is smaller than $n_0$ and the second one is greater. On the control chart, two additional lines are then plotted: upper and lower threshold lines, also called warning limits (UWL and LWL—upper warning limit and lower warning limit). If the point is located between the threshold lines, the next sample is collected from quantity $n_1$, and in case of a point location beyond the threshold line, but still within the control limits, a sample of quantity $n_2$ is collected. Such a procedure is the result of process-stability-level analysis. If the process is stable, research may be done with a larger estimation error, that is, with a smaller number of measurements. If the process shows larger changes of mean level in relation to mean target value, then the estimation error should be cut down, which results in the collection of samples of larger quantity. Control limits of the chart change their location according to the quantity of currently collected samples and adapt to their level, which is a function of the current value of diagnosed property, the variability distribution of which does not have to be normal [17].

### 2.2. Control Charts with Variable Sampling Periods

Nowadays, the following names of control charts based on sampling periods are accepted:

FSI—fixed sampling intervals;

VSI—variable sampling intervals.

In the traditional approach, a control chart is created based on samples collected in constant, predetermined time periods. For today's production processes, such a type of control is not sufficient. Therefore, the use of a variable sample collection period is reasonable. If points on the control chart are approaching intervention limits, it might be that subsequent points will step out beyond control limits, displaying the range of natural process variability. It is important to quickly detect changes in the process and take another sample instead of waiting for another sample from the normal round. If points move close to the central line, the use of a longer sample collection period than the standard one may be favorable.

Summarizing, the proposed control process is based on the determination of the time period for next sample collection depending on current sample values. The time period will be shorter if the process is supposed to be out of control, and longer if such supposition is absent. Issues around the determination of a sampling schedule with variable intervals were considered by researchers whose works are grouped in the literature [11–13]. The problem discussed in this article relates to a complex and multi-step technological process [18–23].

Let us assume that the control chart with a variable sampling interval uses a finite number of time intervals $d_1, d_2, \ldots, d_n$ (Figure 1) expressed in minutes, where $d_1 < d_2 < \ldots < d_n$.

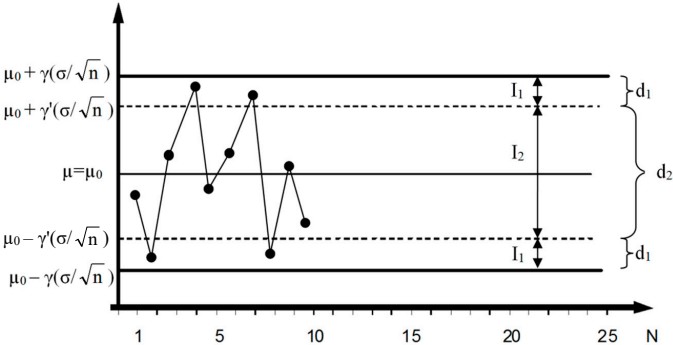

**Figure 1.** $\overline{x}$ control chart scheme with variable sampling intervals (source: based on [12]); N—sample number, ordinate axis—mean values of the samples, $\overline{x}$—mean value of the sample.

Figure 1 presents the principle and architecture of a variable sampling interval control chart. On the chart, two frequency ranges are marked: $I_1$ (Equation (1)) and $I_2$ (Equation (2)), for two used time intervals between sample collecting $d_1$ and $d_2$. Time intervals are determined as follows (Figure 1):

$$I_1 = \left[\mu_0 - \gamma(\sigma/\sqrt{n}),\ \mu_0 - \gamma'(\sigma/\sqrt{n}\right] \cup \left[\mu_0 + \gamma'(\sigma/\sqrt{n}),\ \mu_0 + \gamma(\sigma/\sqrt{n})\right] \qquad (1)$$

$$I_2 = \left[\mu_0 - \gamma'(\sigma/\sqrt{n}),\ \mu_0 + \gamma'(\sigma/\sqrt{n})\right] \qquad (2)$$

where:

μ—process mean (average of all measurement results),
$\mu_0$—process expected value (e.g., the middle of the dimension tolerance range),
σ—process standard deviation,
n—sample size,
$\gamma, \gamma'$—standard deviation multiplications.

Let us assume that for the chart, $\gamma = 3$ and $\gamma' = 1$. Then $\mu = \mu_0$, $P(\overline{x} \in I_1) = 0.3146$ and $P(\overline{x} \in I_2) = 0.6827$, which imply that the longer interval was used about two times more often than the shorter one. If we assume that μ moves to $\mu_1 = \mu_0 + 2\sigma/\sqrt{n}$, and that in such a case $P(\overline{x} \in I_1) = 0.6840$ and $P(\overline{x} \in I_2) = 0.1573$, then the shorter interval will be used more often than the longer one. When the shorter interval is used during changes, the frequency of sampling is increased and the time required to obtain samples beyond limit values (to determine through the signal that the process is out of control) is significantly shorter.

In Figure 1, mean sample values are presented in relation to sample number. In practice, it would be necessary to note sample collection times, because constant distances between points on the chart hide the fact that time intervals between points are not equal. For example, the interval between points 1 and 2 is $d_2$, and between points 2 and 3 is $d_1$. The alternative way to build the charts would be to present time on the horizontal axis. However, this would be disadvantageous in the case of significant time differences between subsequent sample collections, because wide empty areas would appear on the control chart that would spoil plot readability and its interpretation possibilities.

*2.3. Properties of Variable Sampling Interval Charts*

The property of a control chart is the time required for the occurrence of the signal indicating that the process is out of control. If the process is stable (it runs within warning limits), then the time between the collection of subsequent samples should be longer, which at the same time implies a smaller number of false alerts; however, if changes to the process mean values are significant, then the time interval between sample collections should be shortened to increase the probability of quick detection of process destabilization.

The number of samples collected until signal occurrence is known as the average run length (ARL). For a constant sampling interval, the ARL may be easily changed to signal occurrence expected time by multiplication of the currently collected number of samples and the determined interval. Moreover, sampling speed will be constant independently of the process mean. But for a chart with a variable sampling interval, time to signal occurrence is not a simple product of sample numbers and the time interval between them, and sampling speed depends on the dynamics of alternation. Therefore, on variable sampling interval charts, there should be traces of both the number of samples until signal occurrence and the time between them.

Below is a comparative analysis of a constant sampling interval control chart and a variable sampling interval control chart. To achieve this, a program in the MatLab environment (which operates both chart types) was elaborated [14]. The program plots control chart x and R. Calculations performed by using charts are based on the algorithm presented in [12] and on the authors' experiences. Because of the large complexity of the calculations, they will not be presented in this paper. The program allows the user to set parameters manually on his own. Their variability will be used in the comparative analysis

of these two charts. The values of control limits and mean sample values are calculated automatically based on historical data.

The main comparative criterion in the program is the number of samples and time spent till occurrence of the signal that the process is out of control. To increase effectiveness of the charts' work, rules describing specific configurations of point locations regarded as the signal that the process is out of control on charts were implemented in the program. Detailed rule descriptions are presented in the author's work [24]. The user can select rules to control the process.

### 3. Process Analysis

The main criterion for the effectiveness assessment of a certain chart is the number of samples and time that passed until the occurrence of the signal that the process is out of control. For FSI charts, samples are collected from the production process (for example) every 15 produced units (workpieces), and for VSI charts, the following numbers of time intervals were used:

- $d_1 = 2$, when points plotted on the chart are close to UCL or LCL,
- $d_2 = 5$, when points are in the A area of the chart,
- $d_3 = 10$, when points are in the B area of the chart,
- $d_4 = 15$, when points are in the C area of the chart.

Traditionally accepted definitions of areas between control limits of the control chart are presented in Figure 2 [15,24].

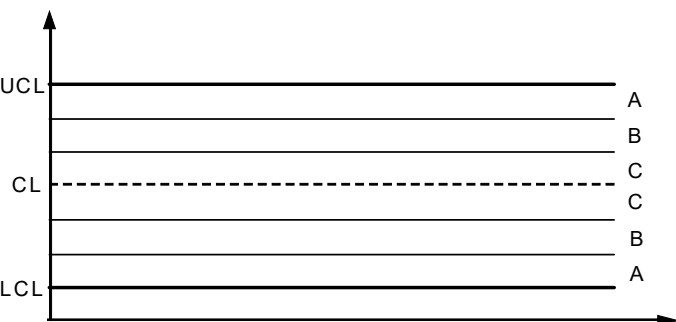

**Figure 2.** Definitions of areas between control limits of the chart.

A production process has been analyzed and its first 60 measurement results are presented in Table 1. The particular production process is not defined, because this is not the essence of the chart effectiveness comparison. Process capability in relation to tolerance limits is also not analyzed. The analyzed subject is process stability in relation to chart control limits, because this is the focus of this paper. Sample size was n = 5. The moment when the process gets out of control was simulated in the program, and it was tested how effectively this will be detected by both types of charts, that is, after how much time and after what number of samples. The analysis results are presented in Table 2, and mean value control charts for the different number of time intervals are presented in Figures 3–5.

**Table 1.** First 60 measurement results used for process assessment (in millimeters).

| | | | | | | | |
|---|---|---|---|---|---|---|---|
| 19.5674 | 20.7143 | 18.8293 | 19.7041 | 19.4118 | 19.8133 | 20.1746 | 20.4173 |
| 18.3344 | 21.6236 | 19.9408 | 18.5249 | 22.1832 | 20.7258 | 19.1846 | 20.3074 |
| 20.1253 | 19.3082 | 18.9894 | 19.7661 | 19.8636 | 19.8133 | 20.3836 | 19.3275 |
| 20.2877 | 20.858 | 20.6145 | 20.1184 | 19.4118 | 20.7258 | 20.1856 | 20.2278 |
| 18.8535 | 21.254 | 20.5077 | 20.3148 | 22.1832 | 19.8133 | 19.1946 | |
| 21.1909 | 18.4063 | 21.6924 | 21.4435 | 19.8636 | 20.7258 | 20.1846 | |
| 21.1892 | 18.559 | 20.5913 | 19.649 | 19.4118 | 19.8133 | 19.8636 | |
| 19.9624 | 20.5711 | 19.3564 | 20.6232 | 22.1832 | 20.7258 | 19.8636 | |

**Table 2.** Measurement results for $\bar{x}$ chart for a different number of time intervals.

| $\bar{x}$ Chart | FSI | VSI | | |
|---|---|---|---|---|
| Number of time intervals | 1 | 2 | 3 | 4 |
| Number of samples till signal occurrence | 70 | 69 | 70 | 82 |
| Time passed till signal occurrence [min.] | 315 | 280 | 280 | 279 |

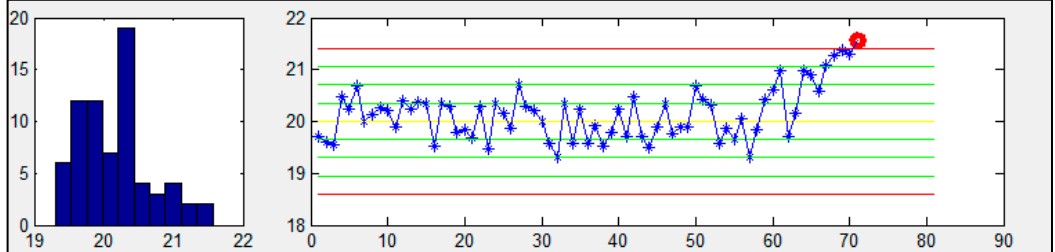

**Figure 3.** $\bar{x}$ control chart with constant sampling interval (FSI).

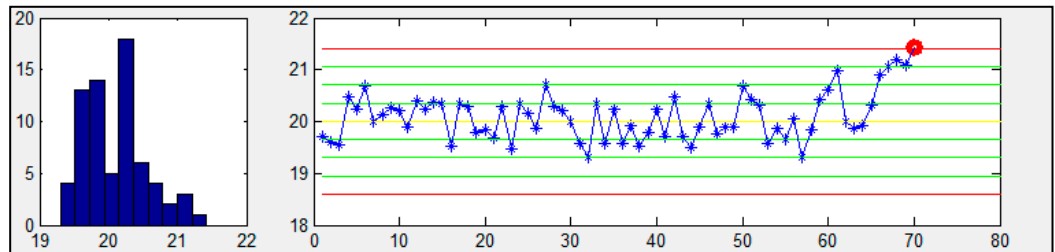

**Figure 4.** $\bar{x}$ control chart with variable sampling interval for two time intervals (VSI).

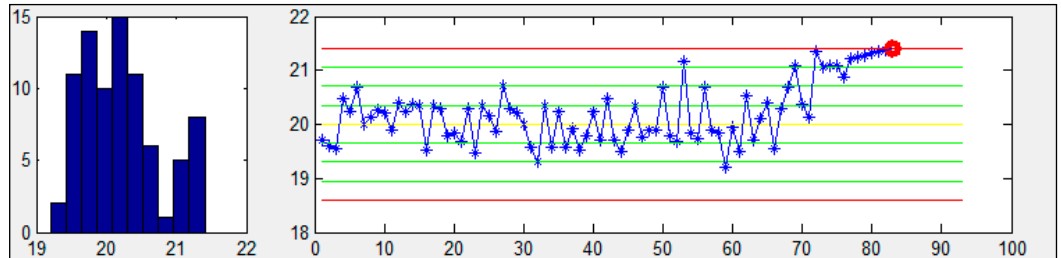

**Figure 5.** $\bar{x}$ control chart with variable sampling interval for four time intervals (VSI).

As can be seen from Table 2, the VSI chart is more effective with two and three time intervals in terms of the number of collected samples when compared with the FSI chart.

The next stage of analysis was devoted to testing the influence of the different sizes of the VSI chart time intervals on detection of the moment when the process gets out of control. Two time intervals, $d_1$ and $d_2$, were used. The calculation results are presented in Table 3, and examples of control charts for certain time intervals are presented in Figures 6 and 7.

**Table 3.** Measurement results for $\bar{x}$ control chart for different sizes of time intervals.

| Interval Size for Chart | | VSI | |
|---|---|---|---|
| $d_1$ [min.] | $d_2$ [min.] | Number of Samples Till Signal Occurrence | Time Passed Till Signal Occurrence [min.] |
| 13 | 15 | 70 | 315 |
| 11 | 15 | 70 | 303 |
| 9 | 15 | 70 | 303 |
| 5 | 15 | 69 | 280 |
| 3 | 15 | 74 | 315 |

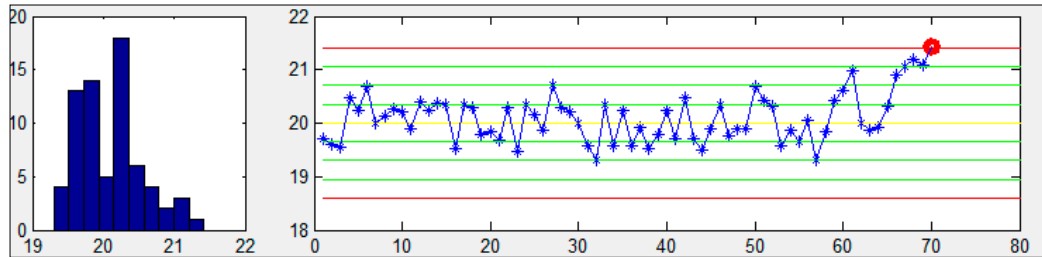

**Figure 6.** An example $\bar{x}$ control chart with variable sampling interval for two time intervals, where $d_1 = 5$, $d_2 = 15$ (VSI).

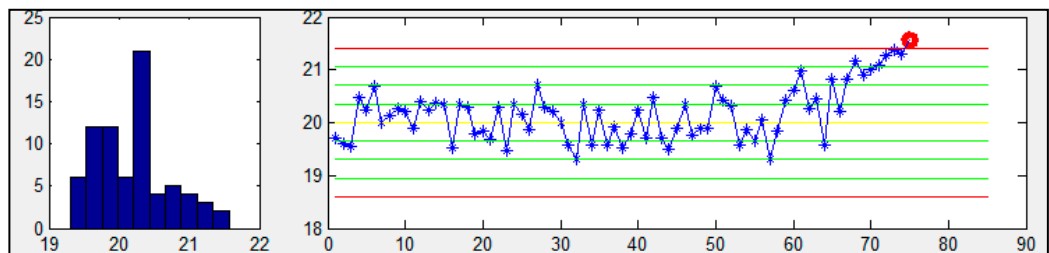

**Figure 7.** An example $\bar{x}$ control chart with variable sampling interval for two time intervals, where $d_1 = 3$, $d_2 = 15$ (VSI).

As can be seen in Table 3, the VSI chart proved to be the most effective for time intervals $d_1 = 5$ and $d_2 = 15$.

In the subsequent step, the influence of different sizes of time intervals on chart effectiveness was tested. The best results for $\bar{x}$ charts were obtained for time interval relations 1:3 (Figure 6). It may be noticed that too large a difference caused aggravation of the results. It is presented in Figures 8–11 how chart effectiveness changed depending on the difference of time intervals.

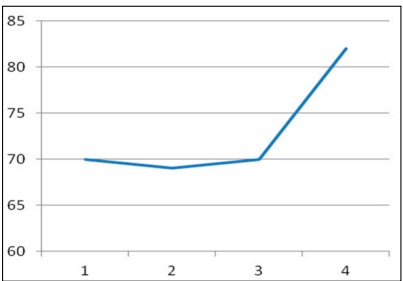

**Figure 8.** Plot of the number of samples till the moment of signal occurrence as a function of the accepted number of time intervals.

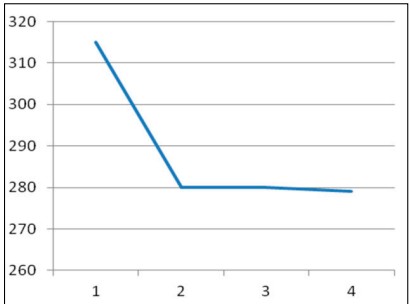

**Figure 9.** Plot of time elapsed till the moment of signal detection in the function of the number of time intervals.

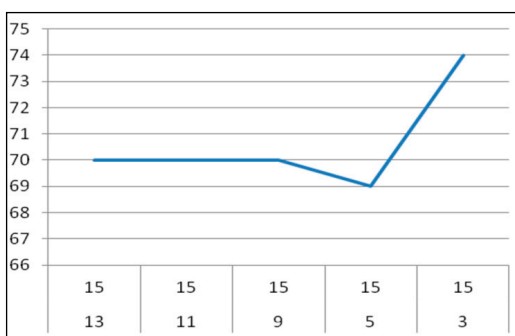

**Figure 10.** Plot of the relation between the number of collected samples till signal occurrence and the size of time intervals.

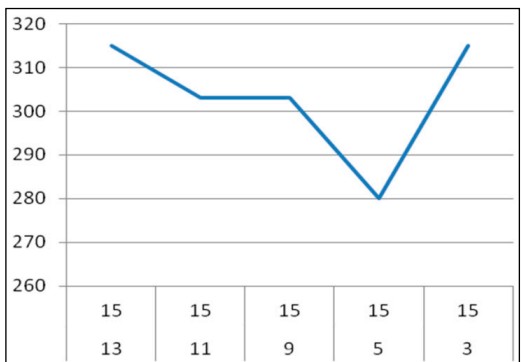

**Figure 11.** Plot of the relation between time till signal occurrence and the difference in sizes of time intervals.

From the presented research, it may be concluded that the VSI chart is more effective than the FSI chart. In the first step, the influence of the number of intervals on chart effectiveness was tested. It can be noticed that too large a number of intervals causes an increase of time till signal occurrence. The study also showed that the most effective results may be obtained using two time intervals with a significant difference between sampling intervals. Too large a number of time intervals did not improve chart effectiveness, but only increased the number of collected samples, which is unfavorable because of the cost of the measurements taken. Before using variable sampling intervals, precise process analysis should be performed to select parameters properly. Results show how important precise adjustment of time intervals is.

During the research, rules for the process getting out of control were used. This increases the effectiveness of both types of charts. In the case of an absence of rules, variable interval charts are less effective than constant interval charts. This part of the research is not presented due to the limited length of the paper.

**Control charts with variable control limits**

During process control, the risk of two types of error should be considered:

- Error of the first type $\alpha$, which means that a false alert occurs although the process was not out of control,
- Error of the second type $\beta$, which means that an alert does not occur although the process was out of control.

The control charts with asymmetrical control limits are used to increase the probability of detection if the process is out of control under the condition that the first error coefficient $\alpha$ has a constant value.

In traditional control charts, it is assumed that the probability of shifts of means from process samples is the same towards positive as towards negative values, which causes acceptance of symmetrical control limits. This assumption is not always correct. In

production practice, there are often situations where the process runs closer to one of the control limits, although no systematic process shift influence was detected. In such cases, the use of symmetrical limits may cause a lack of detection of the signal that the process is out of control or an increase in the number of false alerts.

The essence of control charts with asymmetrical limits is adaptation of the control line location to characteristic process asymmetry. For example, if the frequency of the mean shift towards the lower line is bigger than towards the upper line (Figure 12), then we should expect more false alerts below the lower line. In such a case, the upper control limit is "tightened", that is, moved closer to the central line. As a result, the ability to detect false alerts is the same for both control limits, with the constant value of the first type error, $\alpha$, maintained.

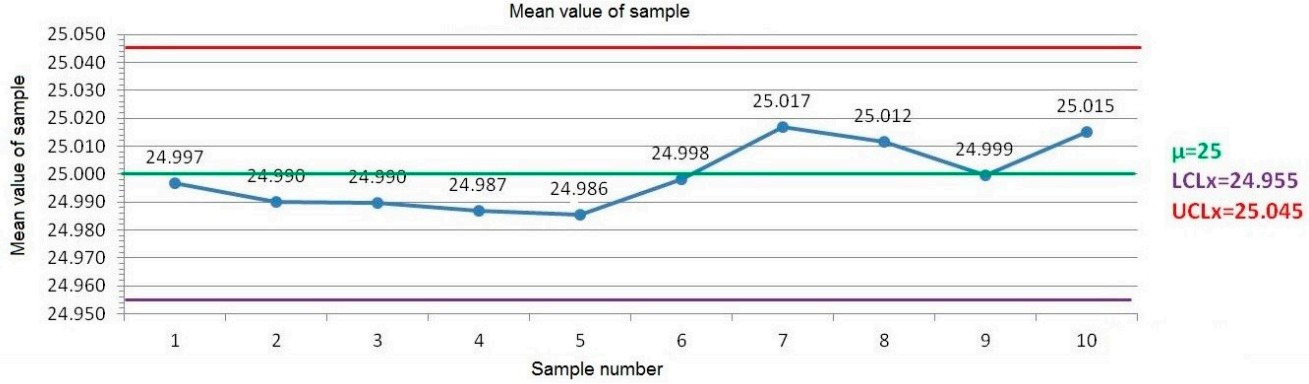

**Figure 12.** Plot of mean values $\overline{x}$ for the process of shaft grinding.

### Example of optimization of control limits for the process of shaft grinding.

The process of shaft grinding with the specification of $25 \pm 0.02$ monitored using the $\overline{x}$-s control chart was analyzed.

Shaft measurement results are grouped in Table 4. For the test, 10 samples were collected (k = 10) with the quantity of 10 ($n$ = 10). Table 5 presents the values of means and standard deviations.

**Table 4.** Measurement results for a series of 100 ground shafts.

| Measur. | Sample 1 | Sample 2 | Sample 3 | Sample 4 | Sample 5 | Sample 6 | Sample 7 | Sample 8 | Sample 9 | Sample 10 |
|---------|----------|----------|----------|----------|----------|----------|----------|----------|----------|-----------|
| 1 | 24.903 | 24.926 | 25.000 | 24.883 | 24.967 | 25.010 | 24.970 | 25.069 | 24.987 | 24.998 |
| 2 | 25.006 | 25.026 | 24.998 | 25.018 | 24.972 | 24.943 | 25.014 | 25.064 | 24.946 | 25.030 |
| 3 | 24.980 | 25.056 | 25.046 | 25.074 | 25.007 | 25.020 | 24.950 | 25.015 | 25.005 | 24.991 |
| 4 | 25.055 | 24.978 | 24.938 | 24.936 | 24.964 | 24.968 | 25.090 | 25.016 | 24.962 | 25.027 |
| 5 | 24.927 | 24.917 | 24.989 | 25.029 | 25.110 | 25.007 | 24.997 | 25.004 | 25.070 | 25.021 |
| 6 | 25.040 | 25.019 | 24.994 | 24.975 | 25.010 | 25.014 | 24.950 | 25.009 | 25.006 | 24.887 |
| 7 | 25.013 | 25.041 | 24.967 | 25.012 | 24.953 | 25.018 | 25.071 | 25.032 | 24.914 | 25.059 |
| 8 | 25.041 | 25.041 | 25.023 | 24.927 | 24.925 | 25.042 | 25.023 | 25.003 | 25.048 | 25.003 |
| 9 | 24.974 | 24.947 | 25.007 | 25.031 | 24.957 | 24.963 | 25.008 | 24.965 | 24.994 | 25.003 |
| 10 | 25.028 | 24.949 | 24.936 | 24.985 | 24.991 | 24.995 | 25.094 | 24.939 | 25.063 | 25.132 |

**Table 5.** Mean value $\overline{x}$ and standard deviation $s$ in the shaft machining process.

| Sample | 1 | 2 | 3 | 4 | 5 | 6 | 7 | 8 | 9 | 10 |
|--------|---|---|---|---|---|---|---|---|---|-----|
| $\overline{x}$ | 24.997 | 24.990 | 24.990 | 24.987 | 24.986 | 24.998 | 25.017 | 25.012 | 25.000 | 25.015 |
| S | 0.051 | 0.052 | 0.035 | 0.058 | 0.051 | 0.031 | 0.054 | 0.040 | 0.051 | 0.061 |

The mean value of means was:

$$\overline{\overline{x}} = \frac{\sum \overline{x}}{k} = 24.999 \tag{3}$$

The $\bar{x}$ control chart is presented in Figure 12 and the standard deviation chart in Figure 13.

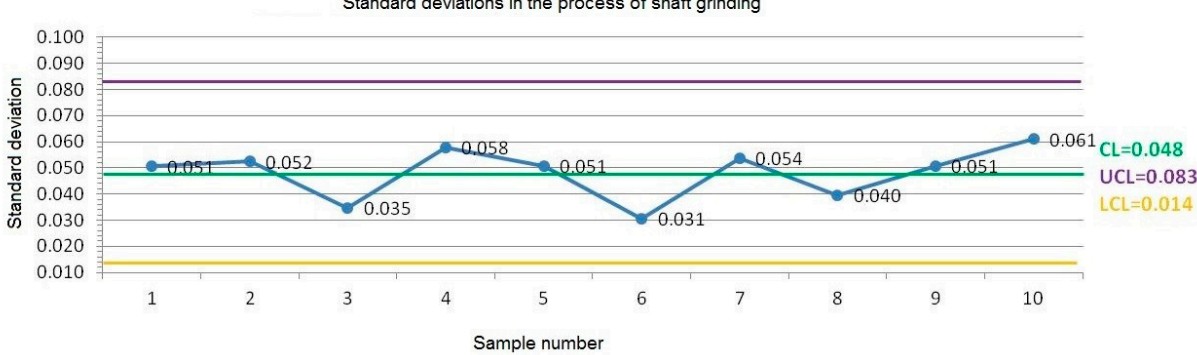

**Figure 13.** Chart of standard deviations in the process of shaft grinding.

As shown in Figures 12 and 13, all points are located close to the central line between the accepted control limits, as well as on the mean chart and on the standard deviation chart. In the case of the mean chart, the number of points located below the central line is bigger than the number of points above the central line, and in the case of the standard deviation chart, this relationship is opposite—the number of points above the central line is bigger than below. In such situations, to control the process properly, optimization of limit locations should be performed. To do this, the algorithm for mean chart optimization described in [21] was used. Its main criterion is to detect false alerts with equal probability for both control limits, while maintaining the constant value of the first type error $\alpha$.

In Figure 14, a chart after setting a new limit location is presented, which takes into account process specificity. According to [16,25–32], charts with a modified limit location are almost three times more effective in the detection of the signal that the process is getting out of control than traditional Shewhart charts.

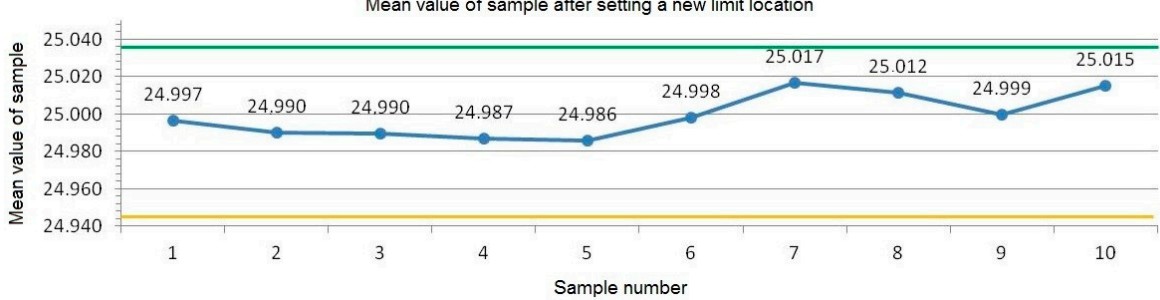

**Figure 14.** Plot of mean sample values $\bar{x}$ in the shaft grinding process after setting new control limits.

## 4. Conclusions

The presented research proved that the use of nonstandard control charts is necessary in many production processes. The use of standard tools such as Shewhart control charts is effective in cases when the process is not influenced by undetectable and unrecognized factors. Process complexity and dynamic process flow circumstances require a reflection of the variability of their parameters such as sample size, collection time intervals and location of control limits on charts. In cases when processes do not flow in a typical way, more sophisticated techniques and tools should be used to protect from false and misleading result interpretations. Nowadays, various types of control charts are used in production practice. They are often chosen improperly and misinterpreted, and so, the aim of further work of the authors will be to design one versatile control chart of a dynamic type that takes process specificity into account, and to facilitate its supervision.

**Author Contributions:** Methodology, T.S.; Validation, T.C.; Formal analysis, J.C. All authors have read and agreed to the published version of the manuscript.

**Funding:** This research received no external funding.

**Data Availability Statement:** Not Applicable.

**Conflicts of Interest:** The authors declare no conflict of interest.

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
