# Peer review of "Statistical Process Control Using Control Charts with Variable Parameters"

_processes, doi:10.3390/pr11092744_

Round 1
Reviewer 1 Report
The paper delves into the quality management in processes, emphasizing the significance of monitoring, diagnosing, and supervising these processes via control charts. While the author provides a comprehensive review of conventional control charts like x-R, x-s, CUSUM, and EWMA, the unique angle of the paper lies in its discussion about atypical control charts which consider variable parameters. This is crucial, given that traditional charts might lead to grave inaccuracies in specific scenarios.
However, while the premise of the paper is intriguing, there are some pertinent issues that need addressing:
Tables and Figures: A glaring discrepancy exists between the tables, figures, and the paper's typesetting.
Bibliography: The list of references feels underwhelming. For a topic of such depth, expanding the bibliography is essential.
Case Study Application: The case study presented is not detailed, on purpose. Iw would indeed be beneficial to have a real example to evaluate better the possibilities provided by the new proposed charts. Integrating a description of the case study would enhance the paper’s practical value, offering readers insights into its tangible benefits.
Equation Numbering: Adding numbering to the equations would make referencing and understanding much more straightforward.
english quality is ok.
Author Response
Thank you for your review. All comments were considered and included in the text of the manuscript. Other notes:
- case study. The reviewer is right that the case study is deliberately not detailed. Control charts for supervising in the shaft machining process are presented. Control cards are dedicated primarily to accurate, difficult to implement and costly processes,
- lines 98-99: P=0.3146 and P=0.6827 are multiples of the standard deviation expressed as probability,
- table 1 shows only the first 60 measurement results due to the limited length of the article.
The area of research related to the supervision of process stability using the control charts technique is still up to date. The problem is the correct selection of the distribution of feature variability (in machine technology - e.g. object dimension) and the control chart. There are many types of charts, such as x-R, x-s, CUSUM, EWMA and others, which are characterized by variable parameters.
The aim of this article was to present the results of research related to control charts with variable parameters. Charts with a variable sample size, variable sampling times and variable location of control limits were considered separately. On the basis of the performed algorithms, software was developed in MatLab, which will make it much easier for the supervisor of the process to decide on corrective and preventive actions.
The further aim of the research will be to develop an algorithm and construction of one control chart taking into account the variability of all its parameters. Soon the authors plan to submit another paper in this area of research. The construction of the chart, in addition to its variable parameters, will take into account the history of the process and sensitivity to large and small changes in the process. The author (T. Sałaciński) has extensive experience in the field of SPC - Statistical Process Control, as evidenced by two of his scientific monographs published in English, and thus having a global reach:
- Sałaciński, T. SPC - Statistical Process Control, The Warsaw University of Technology Publishing House, Warsaw, 2015.
- Sałaciński, T. Quality engineering in manufacturing technology, The Warsaw University of Technology Publishing House, Warsaw, 2019.
Reviewer 2 Report
The manuscript titled “Statistical process control using control charts with variable parameters” reviews the theory behind the use of control charts which use variable numbers of samples, variable periods of sampling, and/or variable control limits.
The organization and language of the paper are reasonable, although both could be improved. The paper has a major shortcoming, it does not make a significant contribution to the literature. Therefore, my recommendation for the paper is "Reject." I justify my recommendation below.
The application of control charts with variable parameters is not novel, on the contrary, it has been around for a very long time. The paper claims to address control chart design, which boils down to selecting parameters used in the control charts. These are simple tasks for someone with a proper background in Statistical Process Control. The paper does a poor job of motivating the study, as well as discussing the outcomes and hence justifying its contribution.
Overall, this referee could not see enough significance in the paper to deserve publication at this level. The authors may want to publish their work in a more practice-oriented journal.
Author Response
The area of research related to the supervision of process stability using the control charts technique is still up to date. The problem is the correct selection of the distribution of feature variability (in machine technology - e.g. object dimension) and the control chart. There are many types of charts, such as x-R, x-s, CUSUM, EWMA and others, which are characterized by variable parameters.
The aim of this article was to present the results of research related to control charts with variable parameters. Charts with a variable sample size, variable sampling times and variable location of control limits were considered separately. On the basis of the performed algorithms, software was developed in MatLab, which will make it much easier for the supervisor of the process to decide on corrective and preventive actions.
The further aim of the research will be to develop an algorithm and construction of one control chart taking into account the variability of all its parameters. Soon the authors plan to submit another paper in this area of research. The construction of the chart, in addition to its variable parameters, will take into account the history of the process and sensitivity to large and small changes in the process. The author (T. Sałaciński) has extensive experience in the field of SPC - Statistical Process Control, as evidenced by two of his scientific monographs published in English, and thus having a global reach:
- Sałaciński, T. SPC - Statistical Process Control, The Warsaw University of Technology Publishing House, Warsaw, 2015.
Reviewer 3 Report
Please, find some comments attached below.

Minor corrections are necessary.
Author Response

(The authors gave the same response as above.)

Round 2
Reviewer 2 Report
Unfortunately, I do not see any improvement in the paper that makes it suitable for publication, in my opinion.
Author Response
I stand by my answer from the previous review:
The aim of this article was to present the results of research related to control charts with variable parameters. Charts with a variable sample size, variable sampling times and variable location of control limits were considered separately. On the basis of the performed algorithms, software was developed in MatLab, which will make it much easier for the supervisor of the process to decide on corrective and preventive actions.
The further aim of the research will be to develop an algorithm and construction of one control chart taking into account the variability of all its parameters. Soon the authors plan to submit another paper in this area of research. The construction of the chart, in addition to its variable parameters, will take into account the history of the process and sensitivity to large and small changes in the process. The author (T. Sałaciński) has extensive experience in the field of SPC - Statistical Process Control, as evidenced by two of his scientific monographs published in English, and thus having a global reach:
- Sałaciński, T. SPC - Statistical Process Control, The Warsaw University of Technology Publishing House, Warsaw, 2015.
- Sałaciński, T. Quality engineering in manufacturing technology, The Warsaw University of Technology Publishing House, Warsaw, 2019.
Reviewer 3 Report
The format of the mathematical relations must be improved in accordance with the standards.
Between minor and moderate editing of English language required.
Author Response
All comments of the Reviewer have been taken into account. In the abstract added:
„Modern control charts are a response to the requirements of Industry 4.0 and are a very good tool in supervising production processes. Their use together with Cp, Cpk indices and other process capability indices is a starting point for process improvement”.
The format of the mathematical relations will be consulted with the publisher.
Round 3
Reviewer 2 Report
The authors clarified the objective of their research.